# A Novel Approach for the Assessment of Cities through Ecosystem Integrity

**Ian MacGregor-Fors** [1,2,*] , **Ina Falfán** [1,3] , **Michelle García-Arroyo** [1,2], **Richard Lemoine-Rodríguez** [4] , **Miguel A. Gómez-Martínez** [5], **Oscar H. Marín-Gómez** [1,6], **Octavio Pérez-Maqueo** [1] and **Miguel Equihua** [1]

1    Red de Ambiente y Sustentabilidad, Instituto de Ecología, A.C. (INECOL), Xalapa 91073, Mexico; ina.falfan@st.ib.unam.mx (I.F.); michelle.garciaarroyo@helsinki.fi (M.G.-A.); oschumar@iztacala.unam.mx (O.H.M.-G.); octavio.maqueo@inecol.mx (O.P.-M.); miguel.equihua@inecol.mx (M.E.)
2    Ecosystems and Environment Research Programme, Faculty of Biological and Environmental Sciences, University of Helsinki, 15140 Lahti, Finland
3    Laboratorio de Restauración Ecológica, Instituto de Biología, Universidad Nacional Autónoma de México, Ciudad de Mexico 04510, Mexico
4    Institute of Geography, Ruhr University Bochum, 44801 Bochum, Germany; Richard.LemoineRodriguez@rub.de
5    Instituto de Biotecnología y Ecología Aplicada (INBIOTECA), Universidad Veracruzana, Xalapa 91090, Mexico; zS18016054@estudiantes.uv.mx
6    Laboratorio de Ecología, Facultad de Estudios Superiores Iztacala, Universidad Nacional Autónoma de México, Tlalnepantla 05490, Estado de México, Mexico
*    Correspondence: ian.macgregor@helsinki.fi

**Abstract:** To tackle urban heterogeneity and complexity, several indices have been proposed, commonly aiming to provide information for decision-makers. In this study, we propose a novel and customizable procedure for quantifying urban ecosystem integrity. Based on a citywide approach, we developed an easy-to-use index that contrasts physical and biological variables of urban ecosystems with a given reference system. The Urban Ecosystem Integrity Index (UEII) is the sum of the averages from the variables that make up its intensity of urbanization and biological components. We applied the UEII in a Mexican tropical city using land surface temperature, built cover, and the richness of native plants and birds. The overall ecosystem integrity of the city, having montane cloud, tropical dry, and temperate forests as reference systems, was low ($-0.34 \pm$ SD 0.32), showing that, beyond its biodiverse greenspace network, the built-up structure highly differs from the ecosystems of reference. The UEII showed to be a flexible and easy-to-calculate tool to evaluate ecosystem integrity for cities, allowing for comparisons between or among cities, as well as the sectors/regions within cities. If used properly, the index could become a useful tool for decision making and resource allocation at a city level.

**Keywords:** citywide; ecological integrity; integrity index; species retention; urban ecology; urbanization intensity

## 1. Introduction

Urbanization leaves some of the most impressive imprints of the human population and activities on ecosystem structure and function [1,2]. Its effects, assessed from an urban-metabolism abstraction, extend far beyond city limits, posing pressing social, environmental, and ecological challenges [1,3]. Undoubtedly, cities have been proven to provide an important set of advantages in terms of economy, governance, and access to services (e.g., electricity, drinking water, health care, education [4]). Yet, cities are highly heterogeneous in many dimensions. Urban lifestyle not only has a profound impact on ecosystems [5–7], but it also extensively affects human health [8]. All the above makes it imperative to revisit the way in which we urbanize landscapes and manage our cities to

build more biodiverse and livable cities, where efforts toward more sustainable practices result in resilient systems.

Urban heterogeneity and complexity have prompted research interest across many disciplines, from architecture, the humanities, and social sciences, to the natural sciences [2,9–12]. To tackle urban complexity, several indices have been proposed [13,14]. Such metrics most commonly aim to provide components to decision makers for them to have solid evidence upon which they can base their actions [15–17]. Some indices focus on different aspects of the urban environment and population, such as lifestyle, economy, ecosystem "health", and biodiversity, aiming to assess urban environmental conditions and resilience [18]. Among those indices, two are the most frequently used: the Ecological Footprint and the City Biodiversity Index (also known as the Singapore Index on Cities' Biodiversity). The Ecological Footprint index evaluates "urban sustainability" based on ecological and economic principles and concepts (such as carrying capacity and natural capital) by accounting for the area of biologically productive land and sea necessary to produce the renewable resources the urban population demands and to assimilate the generated waste [7]. On the other hand, the Singapore Index assesses biodiversity at a local level and provides cities with the ability to monitor their conservation efforts over time [19]; it consists of 23 indicators focused on three key components: native biodiversity, ecosystem-services provision, and governance and management [20,21]. However, an important area of opportunity in the approach to urban complexity has been overlooked, which is based on the ecosystem integrity concept [22,23].

Ecosystem integrity (also referred to as ecological integrity, biotic integrity, biological integrity, natural integrity, integrity of the landscape, or integrity of habitats) was initially defined as "the capability of supporting and maintaining a balanced, integrated, adaptive community of organisms having a species composition, diversity, and functional organization comparable to that of natural habitat of the region" [24] (p. 56). This vision encompasses elements such as the health, biodiversity, stability, sustainability, naturalness, wildness, and beauty of the ecosystems [15] in such a way that a system with integrity can withstand and recover from most perturbations, both natural and induced [24]. As an attribute of ecosystems [25], integrity can be measured through the sum of their chemical, physical, and biological integrities [15,24]. It is also considered as a tool for the management and restoration of ecosystems, for species and environmental protection, for providing services [15,25,26] (see Rohwer and Marris [27]), and for being a bridge among scientists, managers, policymakers, and the general audience for decision making [28,29].

Ecosystem integrity can be used from a local to a global level, in a wide array of systems [30–36], although its use has been higher in natural ecosystems than in human-dominated systems [36]. Considering this wide spectrum, it can actually be even more informative in highly disturbed systems if the reference systems are correctly identified [15,26]. In 2004, Noss [23] introduced the first adaptation of the concept of ecosystem integrity and its evaluation for urban systems, conceiving it as a comparative measure between the built-up areas and the surrounding "natural" systems taken as a reference within a biogeographic region. He clearly recognized that "although urban areas will never have the biodiversity, naturalness, and ecological resilience of pristine wilderness areas, there are reasonable standards they can meet" [23] (p. 4).

Bringing ecosystem integrity into the assessment of urban systems provides an opportunity to push our understanding of cities forward in comparison to previous urban metrics. As such, in this study, we propose a novel and customizable approach to quantify the integrity of urban ecosystems. Briefly, based on a citywide approach, we developed an easy-to-use index that takes as many variables as desired from the physical and biological dimensions of urban ecosystems, which, upon proper scaling and standardization, are contrasted with a given reference system. As citywide assessments are required to populate the index, ecosystem integrity values can be retrieved for entire cities—which can be contrasted to that of other cities—as well as sectors/regions within cities. To show how the index performs, we applied it to a Neotropical green city (Xalapa, Mexico), a widely studied urban system, with a set of variables that allowed us to test its use and performance; yet,

as alerted above, the index is customizable and can be calculated with a different set of variables as long as they meet the criteria reviewed in Section 2.1.

## 2. Materials and Methods

### 2.1. Urban Ecosystem Integrity Index

To fully understand urban ecosystems, both positive and negative aspects, as well as the trade-offs between them, they have to be rigorously assessed [18]. The Urban Ecosystem Integrity Index (UEII) we propose here was conceived as a balance between two urban dimensions: the physical (i.e., urbanization intensity component) and the biological (i.e., species richness).

In the UEII, the urban intensity component subtracts from the ecosystem integrity, while the biological component adds to it (Figure 1). Regarding the urban intensity component, variables should summarize urban encroachment over a given site, such as built cover or human activities (measured directly and/or indirectly; e.g., vehicle traffic, anthropogenic noise, and urban heat islands). For the biological component, it is important to consider only the biodiversity that was recorded in the city and is part of the biota of the reference system or systems [23]. Thus, sites that are more similar in terms of reference biodiversity will indicate, at least partially, higher ecological integrity.

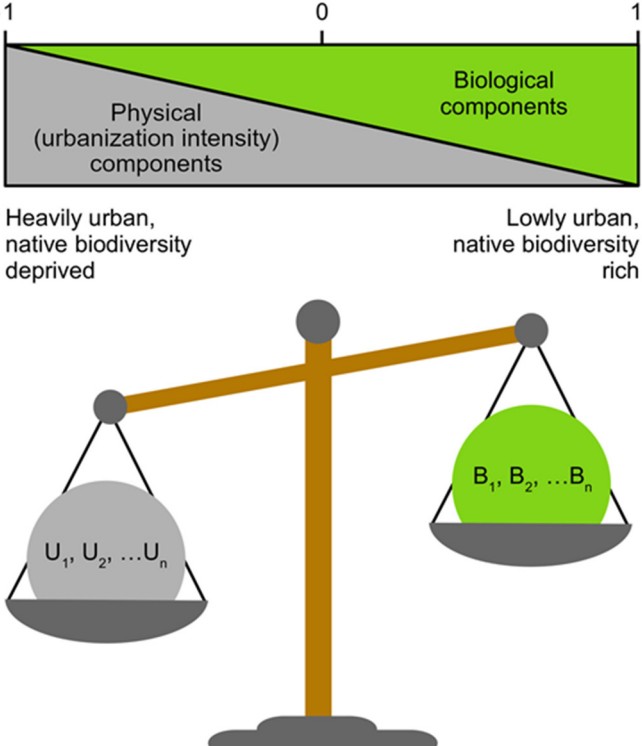

**Figure 1.** Graphical representation of the rationale behind the Urban Ecosystem Integrity Index (UEII), where $U_1$–$U_n$ represent the variables that describe the physical (urbanization intensity) component and pull towards the negative values of the UEII and $B_1$–$B_n$ are the variables describing the biological dimension of the urban continuum, drawing into the positive values of the index, which, in their totality, range from −1 to 1.

As a reference framework, a polygon of the study area is needed as a boundary to perform field surveys and to retrieve remote-sensing data. Although any polygon can be used, we strongly recommend considering the urban continuum, particularly the continuous built-up area of an urban system (see Falfán et al. [37] for an applied example). Mathematically, the UEII is the sum of the averages from all the variables that make up each dimension's component (whose amounts and magnitudes can vary between both

components). Given that the nature of variables can differ greatly, not only conceptually but also mathematically, all variables used in the index are required to be standardized and re-scaled [38].

Finally, the UEII is computed at the pixel level (which needs to be determined a priori, most commonly given by methodological or technological constraints) for an entire urban continuum as follows:

$$UEII = \{[(U_1 + U_2 + \ldots U_n)/n_U] \, (-1)\} + [(B_1 + B_2 + \ldots B_n)/n_B], \tag{1}$$

where $U_1$–$U_n$ represent the variables describing the physical (urbanization intensity) dimension, $B_1$–$B_n$ describe the biological component, $n_U$ is the number of variables used to describe urbanization intensity, and $n_B$ represents that of the biological component. As urbanization components are considered features that erode ecological integrity, they are represented as negative values. On the other hand, biodiversity assets are deemed to be positive contributions. Hence, the range of values of the sum of both components is from –1 to 1 (Figure 2). It is of the utmost importance that the set of variables included to describe the physical and biological dimensions of cities are meaningful and comparable in case that comparative approaches are sought. In the variable selection process, we advise users to consider the potential redundancy (correlation), variance, and previous knowledge of the specific set of chosen variables, to assess crucial aspects of urban systems and to increase the value of UEII results. Although we do not have a suggested list of variables to include for the calculation of the index, we advise that all variables associated with the urbanization intensity component can be used as a proxy (e.g., built cover, temperature, noise, traffic) and those from the biological component follow the "bioindicator" profile (see Moreno et al. [39] for further details). Index values can be used for a single city or for multiple cities. In the case of applying the index in a comparative approach, the variables used to calculate the index ought to be the same across cities, as different proxies could lead to more differences in index values than those actually occurring between (or among) cities.

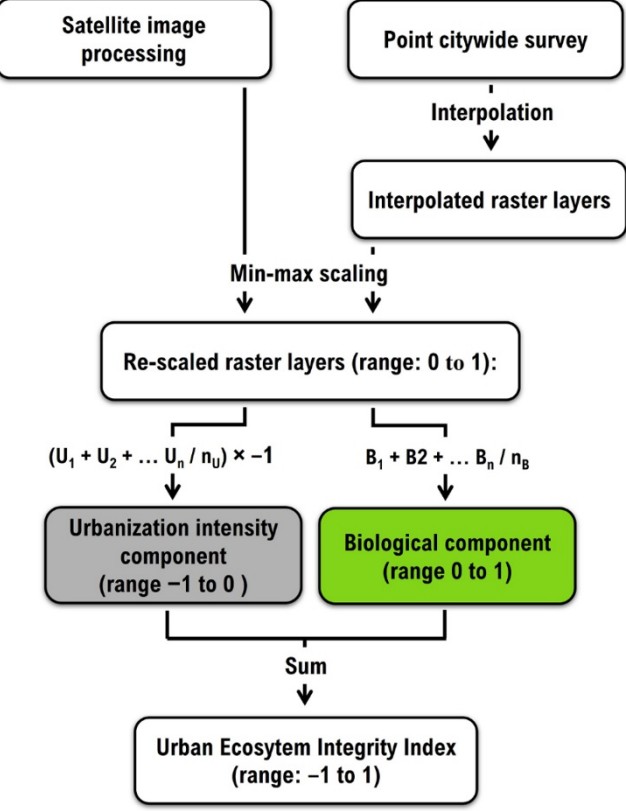

**Figure 2.** Flowchart for the quantification of the Urban Ecosystem Integrity Index (UEII).

*2.2. Case Study: Xalapa*

2.2.1. Study Site

We tested the applicability of the UEII in the Neotropical city of Xalapa, which is located in central Veracruz, Mexico (19°32′38″ N, 96°54′36″ W). The urban continuum of Xalapa (66.53 km$^2$; ~548,282 inhabitants [40]) settles on a transitional climatic region in such a way that it has two main climate types: temperate humid to the northwest and semi-warm humid to the southeast pp. 81-98 in [41].The vegetation of the city was originally comprised of montane cloud, tropical dry, and temperate forests [42], which were used as reference systems for the quantification of the UEII in Xalapa. The city has an important system of public and private vegetation that makes up almost 40% of the city's surface (Falfán et al. [37]). Urban vegetation is composed of local species from these ecosystem types, as well as non-native species from other vegetation types and regions [43,44]. Within its greenspaces and other land uses, Xalapa has been shown to be a city that shelters a sizeable number of wildlife species [45–47].

2.2.2. UEII for Xalapa

To apply the UEII to Xalapa, we firstly delineated its urban continuum. For this, we generated an updated polygon that encompassed the aggregated built-up infrastructure of the city in 2020, following the clustering and communication criteria specified by MacGregor-Fors [48] and Lemoine-Rodríguez et al. [49]. Then, we generated spatially explicit data for the four components that would be used to compute the UEII: land surface temperature (LST) and built cover for the urbanization intensity component and native plant and bird species richness for the biological component.

We retrieved LST (°C) from a Landsat 8 OLI/TIR satellite image with a spatial resolution of 30 m and no cloud cover over the city area (12 April 2019). We pre-processed the image by applying the dark-object subtraction (DOS) image-based atmospheric correction to the visible and infrared bands in order to obtain surface-reflectance values [50], employing the RStoolbox R package [51]. We conducted further processing steps to obtain LST by means of the R package LSTtools [52], used to process thermal data. We converted the digital numbers (DN) of the thermal infrared band 10 (TIR) to at-sensor spectral radiance ($L_\lambda$), employing the multiplicative and re-scaling factors contained in the image metadata, as suggested by the USGS [53]. We applied the modified normalized difference vegetation index (NDVI) threshold method [54] to estimate the per-pixel ground emissivity. According to this output, we defined fixed emissivity values for soil, vegetated, and mixed pixels. We derived the brightness temperature (BT) from the at-sensor radiance of the TIR band. After this, we corrected the BT values by using the Planck function to add the influence of the computed per-pixel emissivity and then converted the values from degrees Kelvin to degrees Celsius.

With the aim of extracting the built-up extent inside the city polygon, we employed an atmospherically corrected Sentinel 2 L2A image of 10 m spatial resolution (19 May 2019). We geographically adjusted the Sentinel 2 bands to the Landsat 8 image to make them spatially comparable and snapped the pixel grids based on their upper left corners. We computed the NDVI based on the red and near-infrared (NIR) bands [55]. Applying an NDVI threshold of 0.40, we removed the vegetation pixels. We manually excluded bare-soil and water pixels in order to retain only built-up features. Finally, we computed the total area in m$^2$ of the 10 m built-up pixels inside each 30 m LST pixel.

In the case of the biological components, we followed a standardized citywide sampling scheme [56] to describe native woody plants (i.e., trees and shrubs) and bird species richness across the city (114 sampling sites within 50 m radii of the sites). For the bird surveys, we performed one 10 min point count per site, which occurred from sunrise to four hours later, recording all birds seen or heard at each survey site (except overflying individuals; following Ralph et al. [57]). Surveys took place in April and May 2019 and again in January and February 2020 to encompass the bird species richness of both the breeding and wintering seasons. For the plant surveys, we recorded all woody species

in street verges, median strips, and residential yards (only when visible from the streets in the latter case) located within 50 m of sampling sites during April and May 2019. We recorded as many species as possible in a period of ten minutes. When we were not able to identify species at the site, we collected a sample to later identify in the XAL Herbarium of Instituto de Ecología, A. C. We confirmed species identification, nomenclature, and distribution in Plants of the World Online (POWO [58]). Importantly, not all plants and birds were considered, but only those that would be found in the original regional ecosystems. Given that the biological data were based on a site-specific sampling scheme, we generated interpolated raster layers for both groups through the inverse distance weighting (IDW) interpolation method [59,60].

In order to have comparable data to populate the index, we re-scaled all four raster layers so that values ranged between 0 and 1. For this, we used the Min–Max transformation formula [38] as follows:

$$X_{\text{re-scaled}} = [(X - X_{\text{min}})/(X_{\text{max}} - X_{\text{min}})] \qquad (2)$$

Afterwards, we simply computed Formula (1) provided in Section 2.1 for the index, which was calculated for each $30 \times 30$ m pixel, as follows:

$$\text{UEII}_{\text{Xalapa}} = \{[(\text{LST}_{\text{re-scaled}} + \text{Built cover}_{\text{re-scaled}})/2]\,(-1)\} + [(\text{Native plant richness}_{\text{re-scaled}} + \text{Native bird richnes}_{\text{sre-scaled}})/2] \qquad (3)$$

As aforementioned, pixels can be of any size for the UEII to work, but, in this case, the restriction was the LST, for which we could only retrieve values at that resolution. Thus, we geographically adjusted all layers to be spatially consistent (aligned) at that spatial resolution. For the computation of the input variables, as well as the UEII index, we used R [61] and QGIS 3.4 [62].

## 3. Results and Discussion

The computation of the UEII for Xalapa was informative, highlighting the urban ecology and biodiversity knowledge available to date [41,44,63–66]. The data gathered documented few variables, but were enough to assess the ecosystem integrity of the city. The average UEII for the city was $-0.34$ ($\pm$SD 0.32), showing that, although the city has been considered green and biodiverse outside its greenspace network, an important proportion of the city's sprawl highly differs from the reference ecosystem, as it is heavily urbanized (Figure 3). This can be clearly seen when contrasted with the vegetation cover in Xalapa (previously published by Falfán et al. [37] (p. 16)), where not all greenspaces have the same UEII value. Simultaneously, many of the streetscapes in the south are not fully covered by vegetation, but their UEII values are higher than many streetscapes in the northeastern region of the city.

The graphical representation of the UEII of Xalapa clearly shows higher ecosystem integrity along the greenspace network of the city, while the intermediate values mainly corresponded to built-up areas mixed with public and private greenspaces, including tree-lined streets. Finally, the lower ecosystem integrity values corresponded to densely built zones with sparse and scarce presence of vegetated areas and some exotic and/or invasive birds (Figure 3).

These results concur with previous findings for the city regarding its "gray" (highly urban) and "green" spaces. Actually, the greenspace network of Xalapa has been shown to retain an important part of the native plants and birds from montane cloud, tropical dry, and temperate forests, as well as other groups, such as butterflies, ants, and fungi, which could be further integrated into the UEII calculation [46,47,65,67]. Spatially, the south and the west of Xalapa showed higher values of ecosystem integrity since these zones have more, larger, and higher-quality greenspaces; this region of the city is considered to be the wealthiest one. The center and northeast showed lower integrity values, albeit with some dispersed spaces and streetscapes of higher integrity (Figure 3).

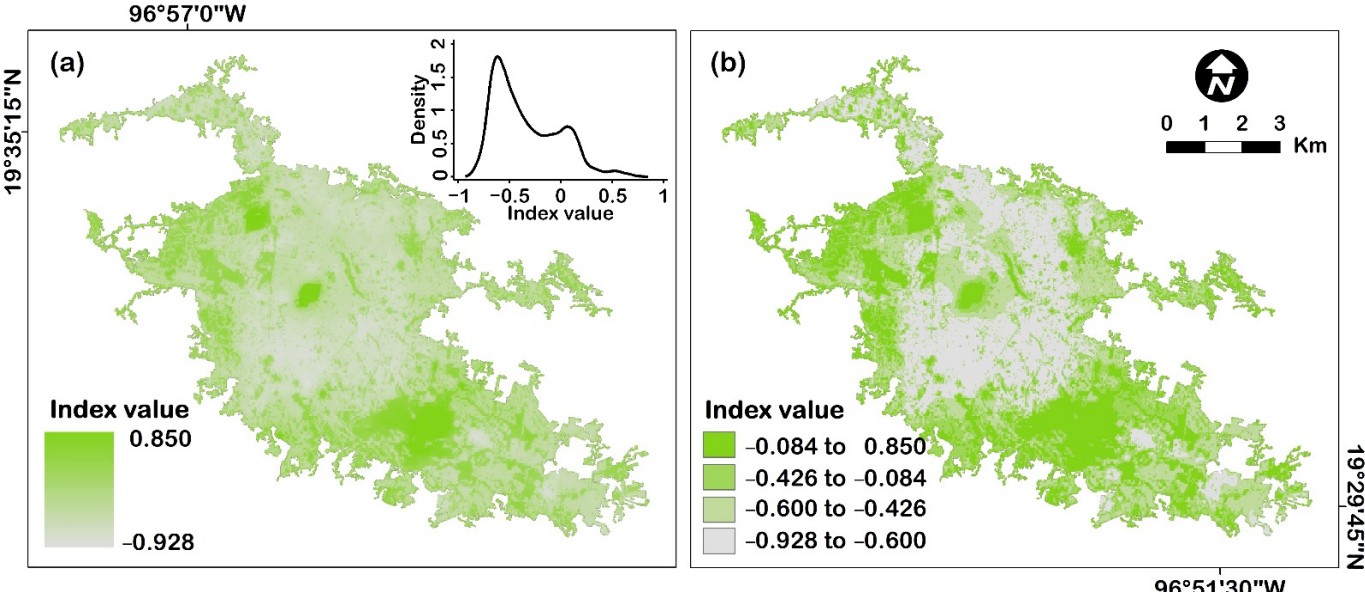

**Figure 3.** Spatially explicit representation of the Urban Ecosystem Integrity Index (UEII) in Xalapa (30 m × 30 m pixel resolution), showing (**a**) continuous results illustrating the distribution of the index values in a density plot and (**b**) its zoning by quartiles.

In the process of computing the index for Xalapa, we identified advantages and limitations, some of which stand out in relation to existent indices, including those focused on urban biodiversity [15,18,21,29]. Among the novel advantages of the UEII is its potential of assessing urban continuum without the constraints of administrative limits. Furthermore, as it is calculated in a spatially explicit way (at the pixel level), the UEII (1) deems cities to be heterogeneous in land cover and land use [5] and (2) can be obtained for all the administrative zones through which the urban continuum extends. Thus, it allows for intra-city comparisons that cannot be achieved with most existing indices since they are calculated considering the city as one homogeneous indivisible entity (e.g., Shaty and Reza [29]). Because the UEII is context-dependent (i.e., the index is a comparative measure with ecosystems of reference), it can be applied to any urban area worldwide, and the number of variables integrated into the index is decided by the researchers based on their knowledge of the cities under study and their intra-environment and surrounding ecosystems. Although previous studies have shown the type of variables suitable for the evaluation of the integrity of urban areas [29], for this index, there is not a fixed number of variables to fulfill. Hence, users can include as many variables as they consider appropriate for their city's conditions. In this study we used four variables—two for each component— which were sufficient for a first evaluation of the index itself and the ecosystem integrity of the urban continuum of Xalapa. When following comparative citywide approaches, UEII values can be contrasted by comparing their corresponding histograms (or density plots; as shown in Figure 3), as long as all variables are the same and were retrieved using similar standardized procedures.

One potential limitation for the calculation of the index is that input information must be spatially explicit. Nonetheless, in most cases, suitable data (apart from the biological component) exist at municipal, regional, or national scales. This aspect could represent a drawback if the biological component information is not available and needs to be generated, and if satellite imagery needs to be generated as well. Yet, there are increasing free options for both cases, such as citizen-based data and free imagery (e.g., remote-sensing information for academic purposes is now available at low or no cost provided by sources such as Google Earth Engine or the United States Geological Survey).

Furthermore, although we only used terrestrial variables in this assessment, variables regarding urban waterbodies (e.g., lakes, ponds, rivers, and dams, both natural and arti-

ficial) may also be included in the UEII since many cities worldwide have an important number of "bluespaces" within their limits. Such urban bluespaces can also be used as, or compared to, reference systems and inform on ecosystem integrity [24,30,34,68,69]. Notably, we did not integrate chemical, socioeconomic, or other physical variables as in other studies (e.g., Shaty and Reza [29]). However, these variables could be included within the urbanization intensity component. The addition of variables to either of the components of the UEII must be clearly specified to express the addition to or subtraction of ecosystem integrity. Additionally, since the UEII was conceived as a measure of ecosystem integrity across the urban continuum, it does not offer information about the effect of the city's metabolism on external ecosystems [7,18], and, therefore, is limited to its boundaries.

Other emerging features of the use of the UEII remain to be explored, such as its contribution to the evaluation of other aspects, such as: ecosystem services, management, conservation, or sustainability measures [15,18,22]; the pertinence of including other wildlife groups, physical variables, or abiotic measures [15,70]; or the validity and usefulness of comparing UEII results with either the same or different reference ecosystems, as has also been recommended for other indices [15,71].

## 4. Conclusions

By computing the UEII with the associated physical variables for the urbanization intensity dimension and biological component, we showed that the ecological integrity of Xalapa could be successfully measured when contrasted with its reference ecosystems. In this example, overall ecosystem integrity, considering montane cloud, tropical dry, and temperate forests as reference systems, was low. Nonetheless, the main greenspaces showed the highest values of ecosystem integrity, as expected.

In general, the UEII showed to be a flexible and easy-to-calculate tool to evaluate urban ecosystem integrity. Thus, the index can provide both basic research and applied management and decision-making uses. On the one hand, colleagues interested in the "ecology of the city" paradigm—aiming at understanding the complexity of entire cities—could use this approach to test hypotheses that require spatially explicit data. Index results can also be used as a metric of urban ecosystem integrity, interpreted according to the input variables. This quantitative metric can then be used as a predictor of dependent ecological variables of interest, falling within the "ecology in the city" paradigm. On the other hand, results of the index could be used for city-scale decision making and resource allocation in developing more sustainable, biodiverse, resilient, and livable cities, following the "ecology for the city" paradigm, seeking to apply the available evidence-based knowledge to action [72,73].

**Author Contributions:** Conceptualization, I.M.-F., O.P.-M. and M.E.; methodology, I.M.-F., I.F. and M.G.-A.; software, R.L.-R.; validation, I.M.-F. and R.L.-R.; formal analysis, I.F.; investigation, I.F., M.G.-A., R.L.-R., M.A.G.-M. and O.H.M.-G.; data curation, I.F. and R.L.-R.; writing—original draft preparation, I.M.-F., I.F., M.G.-A., R.L.-R., M.A.G.-M. and O.H.M.-G.; writing—review and editing, I.M.-F., I.F., M.G.-A., R.L.-R., M.A.G.-M., O.H.M.-G., O.P.-M. and M.E.; visualization, I.M.-F. and I.F.; supervision, I.M.-F.; project administration, I.M.-F.; funding acquisition, O.P.-M. and M.E. All authors have read and agreed to the published version of the manuscript.

**Funding:** This study was supported by Consejo Nacional de Ciencia y Tecnología (CONACYT) through a FORDECYT project: 296842.

**Data Availability Statement:** Data available on request.

**Acknowledgments:** We are most grateful to Eleanor Diamant for her comments and for proofreading the English grammar and wording. We also thank Miguel Á. Domínguez-López for his support with the plant surveys and plant identification, Carlos Durán-Espinosa for plant identification in Herbarium, and Rafael Rueda-Hernández for leading one of the bird survey seasons.

**Conflicts of Interest:** The authors declare no conflict of interest.

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
