# Peer review of "A Novel Approach for the Assessment of Cities through Ecosystem Integrity"

_land, doi:10.3390/land11010003_

Round 1

Reviewer 1 Report

MacGregor-Fors and colleagues created the Urban Ecosystem Integrity Index, which is the sum of the averages of the variables that make up the intensity of urbanisation and the biological component, as described in this paper. They applied this novel index in a Mexican tropical city, using land surface temperature, built cover, and the richness of native plants and birds. The article is well-written and presents a novel approach to quantifying the integrity of urban ecosystems that will attract the interest of the international scientific community.

I only have a couple of comments: 1) the research gap needs to be highlighted in the introduction, 2) the conclusions need more future recommendations.

Author Response

MacGregor-Fors and colleagues created the Urban Ecosystem Integrity Index, which is the sum of the averages of the variables that make up the intensity of urbanisation and the biological component, as described in this paper. They applied this novel index in a Mexican tropical city, using land surface temperature, built cover, and the richness of native plants and birds. The article is well-written and presents a novel approach to quantifying the integrity of urban ecosystems that will attract the interest of the international scientific community.

We greatly appreciate the positive feedback from the reviewer.

I only have a couple of comments: (1) the research gap needs to be highlighted in the introduction

We appreciate the reviewer’s comment. Although we have aimed to develop in such point in L48-67, we have added a phrase in the last paragraph of the introduction to leave this point clearer, as follows:

Bringing ecosystem integrity into the assessment of urban systems provides an opportunity to push our understanding of cities forward in comparison to previous urban metrics. As such, in this study we propose a novel and customizable approach to quantify the integrity of urban ecosystems.”

(2) the conclusions need more future recommendations.

Based on this suggestion, which is similar to that of other reviewers, we have including the following text in the conclusions section:

Thus, the index can provide both basic research and applied management and decision making uses. On the one hand, those interested in the “ecology of the city” paradigm –aiming at understanding the complexity of entire cities– could use this approach to test hypotheses that require spatially-explicit data. Index results can also be used as a metric of urban ecosystem integrity, interpreted according to the input variables. This quantitative metric can then be used as predictors of dependent ecological variables of interest, falling within the “ecology in the city” paradigm. On the other hand, results of the index could be used for city-scale decision making and resource allocation in developing more sustainable, biodiverse, resilient, and livable cities, following the “ecology for the city” paradigm, seeking to apply the available evidence-based knowledge to action [62].”

Reviewer 2 Report

The article addresses the issue of assessing the ecosystem integrity of cities. It declares this in the title of the novel approach. However, the benefits and reasons for using this approach are not sufficiently described and emphasized in the text. What new information does the UEII bring if the resulting rasters de facto overlap with the spread of greenery in the city? For example, how would evaluating the distribution of non-native vegetation affect UEII?
The article is well structured and clearly not written. The methodology presents the indicators used in sufficient detail. In the discussion, the authors correctly address the non-use of other characteristics of ecosystems and their impact on the value of EUII.
Nevertheless, in order to better interpret the significance of the UEII assessment, it would be good to also include a map of urban greenery in the article and to emphasize why the resulting UEII map is more informative.
It would also be appropriate to use a more colorful color range for the image than just shades of green.
Although the article is relatively short, especially in the Conclusion chapter, it brings interesting results. However, their practical significance needs to be emphasized.

Author Response

The article addresses the issue of assessing the ecosystem integrity of cities. It declares this in the title of the novel approach.

We appreciate the kind words and constructive critiques to our paper.

However, the benefits and reasons for using this approach are not sufficiently described and emphasized in the text. What new information does the UEII bring if the resulting rasters de facto overlap with the spread of greenery in the city? For example, how would evaluating the distribution of non-native vegetation affect UEII?

As we state in the introduction: “an easy-to-use index that takes as many variables as desired from the physical and biological dimensions of urban ecosystems, which, upon proper scaling and standardization, are contrasted with a given reference system.” Thus, for the example that the reviewer provides, non-native vegetation could be a measure of difference in relation to native vegetation, which may provide interesting information if that was the intention of the user. We have left crystal clear that our example from Xalapa, with those variables, is an example of use of the UEII, as follows:

To show how the index performs, we applied it to a Neotropical green city (Xalapa, Mexico), a widely studied urban system, with a set of variables that allowed us to test its use and performance; yet, as alerted above, the index is customizable and can be calculated with a different set of variables as long as they meet the criteria (see section 2.1 and Figure 2 for further methodological details).”

Based on this and other comments, we have also included further justification of the need of an index like the UEII.

The article is well structured and clearly not written. The methodology presents the indicators used in sufficient detail. In the discussion, the authors correctly address the non-use of other characteristics of ecosystems and their impact on the value of EUII.

We appreciate the positive feedback.

Nevertheless, in order to better interpret the significance of the UEII assessment, it would be good to also include a map of urban greenery in the article and to emphasize why the resulting UEII map is more informative.

The map of greenery of the city was already published in the past. We believe that presenting that figure is not necessary, but the reviewer rises an interesting point, which we have tried to cover in the following amendment in the Results and Discussion section, properly citing the green cover map:

This can be clearly seen when contrasted with the vegetation cover in Xalapa (previously published by Falfán et al. [2018], p. 16), where not all greenspaces have the same UEII value. Simultaneously, many of the streetscapes in the south are not fully covered by vegetation but their UEII values are higher than many streetscapes of the northeastern region of the city.

It would also be appropriate to use a more colorful color range for the image than just shades of green.

We consider that this is a matter of style and we consider that given the matching colors with figure 1, it is the best way to go. Yet, we leave this decision to the editors.

Although the article is relatively short, especially in the Conclusion chapter, it brings interesting results. However, their practical significance needs to be emphasized.

Based on this comment, together with that of other reviewers, we have included more information in the conclusions, mostly on uses and recommendations.

Reviewer 3 Report

<General Evaluation>

This manuscript reported the assessment of cities by use of ecosystem integrity with the anthropogenic (urbanization intensity components, UIC) and biological components (BIC) in urban. Integrated approach on urbanization’s issue will give an insight for decision makers to evaluate and assess the city and establish a mission and target to achieve the city for citizen in the world. From this perspective, this ms looked to hopeful and message deliver to readers of Journal Land. However, the deeper analysis among variables can show the auto-correlation and the solution for biotic factors that can possess the different spatial and temporal scales could be further needed.

At the title, the meaning of region-specific approach could be handled with the inclusion of “Mexico”, “Tropic Regions”.

Authors have treated the four variables such as UIC (LST and built cover) and BIC (plant and birds, SR). What is a logical ground to assess the UIC with four variables? I do agree with the easy method to evaluate the green and gray with the application of LST and built cover, but there is an auto-correlation between LST and built cover. Min-Max transformation can reduce the noise of data, but it seems that this method would provide the objective-oriented results. I did not find any validation procedure to test your results.

It is recommended that authors proposed the limitation on the research aquatic systems, and chemical and socio-economic variables.

High evaluation and approach could be given on this ms, but the simple analysis on four variables and logical leap discussion would be a de-merit of this ms.

<Detailed comments>

L 226~289, What is your ground to confirm the assess the EI of Xalapa ?

Author Response

This manuscript reported the assessment of cities by use of ecosystem integrity with the anthropogenic (urbanization intensity components, UIC) and biological components (BIC) in urban. Integrated approach on urbanization’s issue will give an insight for decision makers to evaluate and assess the city and establish a mission and target to achieve the city for citizen in the world. From this perspective, this ms looked to hopeful and message deliver to readers of Journal Land.

We appreciate the positive view of our contribution, which is the result of many years of work and understanding of the urban system.

However, the deeper analysis among variables can show the auto-correlation and the solution for biotic factors that can possess the different spatial and temporal scales could be further needed.

At the title, the meaning of region-specific approach could be handled with the inclusion of “Mexico”, “Tropic Regions”.

We understand the comment, but do not fully agree. This may be related to some lack of clarity in some sections (which other reviewers specifically pointed out). Regarding the title, we do not agree in that we need to specify any region, as the index can we used globally, albeit the example is local. Base on this, we have included the following clarification:

To show how the index performs, we applied it to a Neotropical green city (Xalapa, Mexico), a widely studied urban system, with a set of variables that allowed us to test its use and performance; yet, as alerted above, the index is customizable and can be calculated with a different set of variables as long as they meet the criteria (see section 2.1 and Figure 2 for further methodological details).”

Regarding correlation among variables, that would not be a problem in the index, as they would act similarly. Take that there is an hypothetical variable that is 99% correlated with another, used to explore the biological component. Both variables, being added and then divided between them both, would result in something very similar to using 1 variable. But the reviewer is correct in that we should alert the readers about the importance of variable selection, reason why we have included the following text in section 2.1.:

It is of the utmost importance that the set of variables included to describe the physical and biological dimensions of cities are meaningful and comparable in case comparative approaches are sought. In the variable selection process, we advise users to consider potential redundancy (correlation), variance, and previous knowledge on the specific set of variables chosen, to assess crucial aspects of urban systems and to increase the value of UEII results. Although we do not have a suggested list of variables to include for the calculation of the index, we advise that all variables associated with the urbanization intensity component can be used as a proxy (e.g., built cover, temperature, noise, traffic) and those from the biological component follow the “bioindicator” profile (see Moreno et al. [39] for further detail). Index values can be used for a single city or for multiple cities. In the case of applying the index in a comparative approach, the variables used to calculate the index ought to be the same across cities, as different proxies could lead to more differences in index values than those actually occurring between (or among) cities.”

Authors have treated the four variables such as UIC (LST and built cover) and BIC (plant and birds, SR). What is a logical ground to assess the UIC with four variables? I do agree with the easy method to evaluate the green and gray with the application of LST and built cover, but there is an auto-correlation between LST and built cover. Min-Max transformation can reduce the noise of data, but it seems that this method would provide the objective-oriented results. I did not find any validation procedure to test your results.

We agree with the reviewer, but consider that the index, in general, would not be that affected by correlation. Assuming that two variables are highly correlated (even being proxies), their values would “blend”, so if the users include some “redundant” variables, it would not affect the index that much (although if they include 9 correlated ones and 1 uncorrelated one, the component would be clearly biased by this. For this, we have included a phrase in the text regarding the selection and use of variables:

It is of the utmost importance that the set of variables included to describe the physical and biological dimensions of cities are meaningful and comparable in case comparative approaches are sought. In the variable selection process, we advise users to consider potential redundancy (correlation), variance, and previous knowledge on the specific set of variables chosen, to assess crucial aspects of urban systems and to increase the value of UEII results. Although we do not have a suggested list of variables to include for the calculation of the index, we advise that all variables associated with the urbanization intensity component can be used as a proxy (e.g., built cover, temperature, noise, traffic) and those from the biological component follow the “bioindicator” profile (see Moreno et al. [39] for further detail). Index values can be used for a single city or for multiple cities. In the case of applying the index in a comparative approach, the variables used to calculate the index ought to be the same across cities, as different proxies could lead to more differences in index values than those actually occurring between (or among) cities.”

Now, based on the variables pointed out by the reviewer, although they are correlated, there is important variance, and that variance adds importantly to the description of the “harshness” of urbanity:

Finally, given the nature of our sample size and the effort and funds needed to increase the resolution, we could not perform a validation; yet, we agree that that would have been fabulous to do.

It is recommended that authors proposed the limitation on the research aquatic systems, and chemical and socio-economic variables.

We agree with the reviewer and appreciate the comment.

High evaluation and approach could be given on this ms, but the simple analysis on four variables and logical leap discussion would be a de-merit of this ms.

We agree that using more variables would make further amazing analyses. Yet, we were actually amazed by the detail and information provided with a few variables.

Detailed comments

L 226~289, What is your ground to confirm the assess the EI of Xalapa ?

We understand the reviewer’s concern. We have studied Xalapa for over a decade now, and we were surprised on the level of information provided by the index. Based on this comment and one from another reviewer, we have included a text that give some context on this:

This can be clearly seen when contrasted with the vegetation cover in Xalapa (previously published by Falfán et al. [2018], p. 16), where not all greenspaces have the same UEII value. Simultaneously, many of the streetscapes in the south are not fully covered by vegetation but their UEII values are higher than many streetscapes of the northeastern region of the city.”

Reviewer 4 Report

This work is very interesting and can provide useful information at a glance for urban areas - congratulations.

However I do find some aspects that could be improved:

  • The Materials and Methods section about the UEII (2.1) fails to give enough information to completely comprehend the proposed index. Please deepen into the variables (which are the most convenient, which you definitely be included? what's the minimum desirable number? etc, based on research), as the manuscript is too vague on this matter. I understand that the nature of variables "can vary greatly", but it shouldn't be that vague either when it comes to such a relevant part of the research. Please elaborate a little bit more.
  • On the other hand, one of the main aspects that makes an index useful is the ability to compare different urban areas. How can that be achieved in this case? There should be somewhat of a scale for the obtained index values, unless it is not possible, in which case it should be explained. Visually the interpretation is clear (figure 3), but it would be helpful to define ranges for the numerical index values.
  • The bibliography could use more recent references - the amount of research in this topic has increased during the past few years and it is fundamental to study the most up-to-date research!

Author Response

This work is very interesting and can provide useful information at a glance for urban areas – congratulations.

We very much appreciate the positive general view of the reviewer.

However I do find some aspects that could be improved

We appreciate the time and effort in suggesting ways to make our work better.

The Materials and Methods section about the UEII (2.1) fails to give enough information to completely comprehend the proposed index. Please deepen into the variables (which are the most convenient, which you definitely be included? what's the minimum desirable number? etc, based on research), as the manuscript is too vague on this matter. I understand that the nature of variables "can vary greatly", but it shouldn't be that vague either when it comes to such a relevant part of the research. Please elaborate a little bit more.

Based on this comment and others from the other reviewers, we have included a section regarding this topic:

It is of the utmost importance that the set of variables included to describe the physical and biological dimensions of cities are meaningful and comparable in case comparative approaches are sought. In the variable selection process, we advise users to consider potential redundancy (correlation), variance, and previous knowledge on the specific set of variables chosen, to assess crucial aspects of urban systems and to increase the value of UEII results. Although we do not have a suggested list of variables to include for the calculation of the index, we advise that all variables associated with the urbanization intensity component can be used as a proxy (e.g., built cover, temperature, noise, traffic) and those from the biological component follow the “bioindicator” profile (see Moreno et al. [39] for further detail). Index values can be used for a single city or for multiple cities. In the case of applying the index in a comparative approach, the variables used to calculate the index ought to be the same across cities, as different proxies could lead to more differences in index values than those actually occurring between (or among) cities.”

We hope to have made it clear that there is no “golden rule”, but rather depends on the use that researchers and decision makers will give to the outcome.

On the other hand, one of the main aspects that makes an index useful is the ability to compare different urban areas. How can that be achieved in this case? There should be somewhat of a scale for the obtained index values, unless it is not possible, in which case it should be explained. Visually the interpretation is clear (figure 3), but it would be helpful to define ranges for the numerical index values.

As noted in section 2.1 and the caption of Figure 1, the index ranges from -1 to 1. We hope that the rest of the comment is covered in our reply to the last comment. Regarding Figure 3, there are ranges only in the right panel, as we show quartiles, in the case of the left panel, its the continuous data. We have modified the caption for further clarity:

Figure 3. Spatially explicit representation of the Urban Ecosystem Integrity Index (UEII) in Xalapa (30m × 30 m pixel resolution), showing (a) continuous results illustrating the distribution of the index values in a histogram and (b) its zoning by quartiles (right panel).”

The bibliography could use more recent references - the amount of research in this topic has increased during the past few years and it is fundamental to study the most up-to-date research!

We thank the reviewer for pointing this out. We performed an in depth review once more regarding urban indices with this approach and found little (although we did find a couple of them and cited them properly). We also found some interesting novel information regarding ecological integrity, which are now included in the text.

Round 2

Reviewer 3 Report

Many parts were developed and changed, but I do not agree with the title.

Because, There are many cities at the temperate regions in the world, so, the tropical and Mexico case could not represent the all cities around the worlds. Also, the word of 'novel' is an evaluation by authors, the deletion of 'novel' would be suggested.

So, I would like to suggest the title as "A regional approach for the assessment of cities through ecosystem integrity at the Tropic Cities, Mexico."

If you do not accept this suggestion, please drop me as  reviewer.

Author Response

I disagree with the reviewer regarding the title of the paper for two main reasons: (although there are others that are included across the MS):

  • The title refers to "A novel approach". What is novel is the method, not the example of Xalapa. We consider that this approach is quite novel, as nothing similar has been proposed with this use, and has several scientific and decision making applications.
  • We recognize that cities vary across the globe. That is one of the reasons behind the index being customizable and can be used for cities from across the globe. Actually, we recently obtained enough preliminary data for a boreal city and the index works fantastically. Thus, we do not consider that the title needs to focus on a regional approach. As is clear across the manuscript, we used data for Xalapa to test the UEII, not based the index to fit a tropical city or Xalapa specifically.

Finally, I consider that the final remark of the reviewer is quite unfortunate. Firstly, I consider that the reviewer is truncating the academic process of communication. Second, I believe that providing an ultimatum behind anonymity to drop serving as reviewer for the paper if we don't comply with his view seems far from professional (and even rude). 

Ian MacGregor-Fors